# Circulating MicroRNAs from Serum Exosomes May Serve as a Putative Biomarker in the Diagnosis and Treatment of Patients with Focal Cortical Dysplasia

**DOI:** 10.3390/cells9081867

**Published:** 2020-08-10

**Authors:** Shang-Der Chen, Hsiu-Yung Pan, Jyun-Bin Huang, Xuan-Ping Liu, Jie-Hau Li, Chen-Jui Ho, Meng-Han Tsai, Jenq-Lin Yang, Shu-Fang Chen, Nai-Ching Chen, Yao-Chung Chuang

**Affiliations:** 1Department of Neurology, Kaohsiung Chang Gung Memorial Hospital, Kaohsiung 83301, Taiwan; chensd@adm.cgmh.org.tw (S.-D.C.); ultima1229@gmail.com (C.-J.H.); menghan@cgmh.org.tw (M.-H.T.); fangoe1@yahoo.com.tw (S.-F.C.); naiging@yahoo.com.tw (N.-C.C.); 2Institute for Translation Research in Biomedicine, Kaohsiung Chang Gung Memorial Hospital, Kaohsiung 83301, Taiwan; pphome67@yahoo.com.tw (X.-P.L.); jiehau1060301@gmail.com (J.-H.L.); jyang@adm.cgmh.org.tw (J.-L.Y.); 3College of Medicine, Chang Gung University, Taoyuan 33302, Taiwan; 4Department of Emergency Medicine, Kaohsiung Chang Gung Memorial Hospital, Kaohsiung 83301, Taiwan; fornever@cgmh.org.tw (H.-Y.P.); u9001135@gmail.com (J.-B.H.); 5Department of Neurology, School of Medicine, College of Medicine, Kaohsiung Medical University, Kaohsiung 80708, Taiwan; 6Department of Biological Science, National Sun Yat-sen University, Kaohsiung 80424, Taiwan

**Keywords:** focal cortical dysplasia, microRNA, exosomes, biomarkers, circulation

## Abstract

Focal cortical dysplasia (FCD) is a congenital malformation of cortical development where the cortical neurons located in the brain area fail to migrate in the proper formation. Epilepsy, particularly medically refractory epilepsy, is the most common clinical presentation for all types of FCD. This study aimed to explore the expression change of circulating miRNAs in patients with FCD from serum exosomes. A total of nine patients with FCD and four healthy volunteers were enrolled in this study. The serum exosomes were isolated from the peripheral blood of the subjects. Transmission electron microscopy (TEM) was used to identify the exosomes. Both exosomal markers and neuronal markers were detected by Western blotting analysis to prove that we could obtain central nervous system-derived exosomes from the circulation. The expression profiles of circulating exosomal miRNAs were assessed using next-generation sequencing analysis (NGS). We obtained a total of 107 miRNAs with dominant fold change (>2-fold) from both the annotated 5p-arm and 3p-arm of 2780 mature miRNAs. Based on the integrated platform of HMDD v3.2, miRway DB and DIANA-miRPath v3.0 online tools, and confirmed by MiRBase analysis, four potentially predicted miRNAs from serum exosomes in patients with FCD were identified, including miR194-2-5p, miR15a-5p, miR-132-3p, and miR-145-5p. All four miRNAs presented upregulated expression in patients with FCD compared with controls. Through Kyoto Encyclopedia of Genes and Genomes (KEGG) pathway analysis and pathway category of four target miRNAs, we found eight possible signaling pathways that may be related to FCD. Among them, we suggest that the mTOR signaling pathway, PI3K-Akt signaling pathway, p53 signaling pathway, and cell cycle regulation and TGF-beta signaling pathway are high-risk pathways that play a crucial role in the pathogenesis of FCD and refractory epilepsy. Our results suggest that the circulating miRNAs from exosomes may provide a potential biomarker for diagnostic, prognostic, and therapeutic adjuncts in patients with FCD and refractory epilepsy.

## 1. Introduction

Focal cortical dysplasia (FCD) is a congenital malformation of cortical development where the cortical neurons located in a brain area fail to migrate to the proper formation in utero [1,2]. Historically, FCD has been classified into subtypes (Ia, Ib, IIa, IIb, and III) based on neuropathological features [1,3]. Epilepsy, particularly medically refractory epilepsy, is the most common clinical presentation for all types of FCD [1,4,5]. With the advancement of higher-resolution brain magnetic resonance imaging (MRI), FCD can be identified more easily than in previous years, and clearly linked to the foci of seizure onset [6,7,8]. Whereas both genetic and acquired factors are involved in the pathogenesis of cortical dysplasia, the pathogenesis of FCD remains unclear [1,9]. Recent evidence suggests that FCD arises from de novo somatic mutations during brain development; most of these mutations have been identified in genes encoding regulatory proteins within the mechanistic target of rapamycin (mTOR) pathway, suggesting that aberrant mTOR pathway signaling is a critical mechanism accounting for the histopathological features of some FCD subtypes [1,2,10]. Although mutations in several mTOR pathway genes (such as *MTOR*, *DEPDC5*, and *NPRL3*) may be associated with FCD type IIb [10,11], in most patients with FCD, causative gene mutations have not been identified. Until now, there has been a lack of effective and potential biomarkers to elucidate the pathogenesis and epileptogenesis of FCD.

MicroRNA (miRNA) is a noncoding double-stranded RNA molecule with a length of 20–22 nucleotides; the front sequence has a stem‒loop-like hairpin structure. miRNA is known to have a turnover ability to control mRNA degradation and translation by targeting multiple mRNAs. It has been actively explored in numerous physiological and pathological processes where miRNAs act as post-transcriptional inhibitors of gene expression [12,13]. Diverse miRNA profiles have been identified through high-throughput sequencing in biofluid, human cells, and tissues [14,15]. These small noncoding RNAs may be potential novel disease-specific biomarkers, as well as other RNA species of tRNAs, snoRNAs, and piRNAs [14,16]. Changes in the expression of circulatory miRNAs have been noted in many neurological diseases, including Parkinson’s disease [17,18], Alzheimer’s disease [19], amyotrophic lateral sclerosis [20], ischemic stroke [21], and epilepsy [22,23].

Traditional diagnostic methods for epilepsy are based on a clinical history of neurological and epilepsy, electroencephalography, neuroimaging studies, and the clinical experience of physicians. However, there is no surefire method to avoid the misdiagnosis of epilepsy, to elucidate the pathophysiology of epileptogenesis, and to determine the treatment for epilepsy. Genetic diagnosis, especially whole-exome sequencing, is a powerful diagnostic tool for patients with epilepsy, particularly for patients with early-onset epilepsy of an unknown cause [24]. Recently, evidence supports that miRNAs play a crucial role in gene expression regulation in epilepsy, which provides new putative targets for the control of seizure-induced cell damage and as potential biomarkers of epileptogenesis [22,23,25]. In epilepsy-related diseases, preclinical animal [26,27,28,29] and clinical human studies showed the down- or upregulation of miRNAs, which suggested that miRNAs play an important role in the diagnosis, predicted prognosis, and treatment of epilepsy [25,30] and status epilepticus [31].

Exosomes are small (30–150 nm) membrane vesicles that originate from the endocytosis pathway after the fusion of multivesicular endocytic compartments with the cell membrane and are then released into the extracellular environment [32,33]. Exosomes can carry a great variety of molecules, including proteins, lipids, saccharides, mRNAs, and miRNAs, and the exosomal lipid bilayer protects the genetic information from degradation [32,33,34]. Recent studies point to the bodily fluid-derived exosomes as a potential source of miRNAs, and miRNAs in circulating exosomes have been used as potential diagnostic biomarkers for noninvasive diagnosis of cancer and other diseases [34,35,36,37]. Interestingly, recent evidence revealed that a key characteristic of exosomes is that they may travel between the central nervous system (CNS) and the peripheral circulation [38]. This property may lead to the circulating exosomes serving as biomarkers for CNS disorders [38]. Therefore, the alternation of miRNAs in circulating CNS-derived exosomes may be proposed as putative biomarkers of epileptogenesis and pathogenesis in patients with FCD and refractory epilepsy.

In the present study, we characterized the expression profiles of the miRNAs in patients with FCD from the serum exosomes and validated the hypothesis that the mTOR signaling pathway, PI3K-Akt signaling pathway, and other related pathways participate in pathogenesis and epileptogenesis in patients with FCD.

## 2. Materials and Methods

The present study was conducted at Kaohsiung Chang Gung Memorial Hospital, Kaohsiung, Taiwan. The institutional ethics committee approved the study protocol (No. 201700523A3), and informed written consent was obtained from all the subjects.

### 2.1. Subjects

A total of nine patients (five males and four females) with FCD confirmed by brain MRI were enrolled in the present study through the Epilepsy Outpatient Clinic of Kaohsiung Chang Gung Memorial Hospital. Four healthy volunteers (two males and two females) with unremarkable brain MRI were recruited to serve as normal controls. The clinical information of the patients with FCD were obtained from the clinical records and interventions, including seizure types and semiology, age at seizure onset, current anti-epileptic drug (AED) therapeutic state, findings of electroencephalography, and the results of the brain MRI study. The seizure types and semiology were classified according to the 2017 recommendations of the International League Against Epilepsy (ILAE, Flower Mound, TX, USA) [39].

Fasting peripheral blood was obtained from all subjects at 8:00 a.m. In patients with FCD, the peripheral blood was obtained during an interictal state. Serum samples were separated by centrifugation (2000× *g* for 10 min at 4 °C) and stored at −80 °C after collection to isolate the exosomes.

### 2.2. Isolation of Exosomes and RNA Extraction

Intact exosomes were isolated using the Total Exosome Isolation Kit (from serum) (Cat. 4478360, Invitrogen, Carlsbad, CA, USA), following the manufacturer’s protocol. The appropriate volume of exosome precipitation solution from kit was added to the biofluid, and mixed and refrigerated for 30 min. Resuspended exosome pellets in 100‒500 µL PBS were stored at −80 °C until exosomal RNA extraction. Total RNAs (including miRNAs) in the exosomes were extracted from 17 frozen serum samples using the Total Exosome RNA & Protein Isolation Kit (Cat. 4478545, Invitrogen) according to the manufacturer’s protocol.

### 2.3. Transmission Electron Microscopy for Identification of Exosomes

Freshly isolated exosomes were fixed in neutral buffered 4% (*w*/*v*) paraformaldehyde for 1 h and dropped onto a formvar/carbon-coated grid at room temperature for 20 min to be absorbed and dried. After washing in PBS, the exosomes were fixed with 1% (*w*/*v*) glutaraldehyde for 5 min. Then, 1% (*w*/*v*) aqueous uranyl acetate (5 μL) was applied to the grid for 30 s negative staining of the exosomes. The samples were then embedded in a solution of 0.4% (*w*/*v*) uranyl acetate and 1.8% (*w*/*v*) methylcellulose and incubated on ice for 10 min. The exosomes were visualized with a transmission electron microscope (TEM) (HT7800, Hitachi, Tokyo, Japan) operated at 75 kV, and images were acquired by a digital camera.

### 2.4. Measurement of Particle Size and Distribution of Exosomes

A Zetasizer Nano ZS system (Malvern Instruments, Malvern, UK) with a 633 nm helium–neon laser emitting at an angle of 173 degrees was using for measuring the particle size of the exosomes. Resuspended exosomes were diluted 1:100 in sterile PBS to a total volume of 1 mL and loaded into a disposable cuvette for measurement. Data interpretation was via the software bundled with the apparatus and Microsoft Office Excel 2017 (Microsoft, Redmond, WA, USA).

### 2.5. Western Blot Analysis

In the field of exosome research, CD63, heat shock protein 70 (HSP70), and TSG101 are often used as exosome biomarkers and have been proven to be especially enriched in exosomes [40]. To detect the content of CNS-derived exosomes from our sample, we examined the expressions of neuronal markers: MAG, which can be found in oligodendroglial exosomes [41]; GluR2&3, which have been detected in exosomes released by cultured cortical neurons [42]; and NCAM1, which has been utilized to purify neuron-derived exosomes from the serum of Alzheimer’s disease patients [43], to indicate the inclusion of CNS-derived exosomes in our isolated exosomes.

For the Western blot analysis, the primary antisera used included a mouse monoclonal antibody against TSG101 (1:1000, GTX70255, Abcam, Cambridge, MA, USA), NCAM-1 (1:500, sc-106, Santa Cruz Biotechnology, Santa Cruz, CA, USA); rabbit monoclonal or polyclonal antibody against MAG (1:1000, 9043S, Cell Signaling Technology, Danvers, MA, USA), CD63 (1:1000, ab134045, Abcam), HSP70 (1:1000, Cat. 4876, Cell Signaling Technology, Danvers, MA, USA), GluR 2&3 (1:1000, AB1506, Sigma Aldrich, St. Louis, MO, USA). This was followed by incubation with HRP-conjugated secondary antibodies, goat Anti-Rabbit IgG H&L (1:5000, ab672, Abcam) or Goat Anti-Mouse IgG H&L (1:5000, ab6789, Abcam). Specific antibody‒antigen complexes were detected by an enhanced chemiluminescence western HRP substrate (Merck Millipore, Billerica, MA, USA). The images were obtained by Analytik Jena™ UVP ChemStudio PLUS (Analytik Jena US, Upland, CA, USA).

### 2.6. Next-Generation Sequencing Analysis

The expression profiles of miRNA extracted from the serum exosomes were assessed using next-generation sequencing (NGS) analysis [44]. In brief, the total RNAs in the exosomes were extracted from serum samples using a Total Exosome RNA & Protein Isolation Kit (Invitrogen) according to the instruction manual. The purified RNAs were quantified at OD_260nm_ using an ND-1000 spectrophotometer (NanoDrop Technologies, Wilmington, DE, USA) and qualitatively analyzed using a Bioanalyzer 2100 (Agilent Technologies, Santa Clara, CA, USA) with RNA 6000 LabChip kit (Agilent Technologies). Library preparation and deep sequencing for NGS were carried out at Insight Genomics Inc. (Tainan, Taiwan) as the official protocol of Illumina (San Diego, CA, USA) with the Illumina NextSeq sequencing platform and a Mid Output v2.5 kit (150 cycles) Cat. #20024904 (Illumina). The analysis pipeline was developed by the CLC Genomics workbench v10.1 software package. First, reads shorter than 15 bps and longer than 35 bps with low quality scores were removed from the dataset. The read counts of miRNA were analyzed statistically and represented by reads per kilobase per million (RPKM). Then, the reads were counted on different types of small RNAs in the data and compared to the miRBase database (miRBase v22.1; http://www.mirbase.org/). The small RNAs were annotated with the name of miRNAs. For differential expression analysis, two sets of annotated samples were analyzed by the tests on proportions based on the test of Kal et al., 1999 [45]. Cluster analysis for differentially expressed miRNAs was performed using the Heatmap Mev4.9 software [46].

### 2.7. Analyses Using MicroRNA Target Predicting Databases

We used comprehensive atlases of predicted and validated miRNA‒target interactions, including the Human MicroRNA Disease Database (HMDD; v3.2) (http://www.cuilab.cn/) [47], miRwayDB (http://www.mirway.iitkgp.ac.in/) [48], and the DIANA-miRPath v3.0 online tool (http://snf-515788.vm.okeanos.grnet.gr/) [49], to identify the network components of significantly differentially expressed miRNAs. For further confirmation, MiRBase, an open-source software library providing comprehensive prediction of miRNA targets, was used to predict the potential targets of significantly expressed (>2-fold change) miRNAs related to FCD, seizure, epilepsy, or status epilepticus.

### 2.8. Analysis of Disease-Related Pathways from Predicted MicroRNA and Pathway Category

The Kyoto Encyclopedia of Genes and Genomes (KEGG) database [50] was used in the analysis of disease-related pathways from the significantly expressed miRNAs. All analyses for KEGG pathways and pathway target genes were completed by the DIANA-miRPath v3.0 online tool. We selected pathways with *p*-values less than 0.05.

We utilized two integrated platforms, BioPlanet (https://tripod.nih.gov/bioplanet/#) [51] and the KEGG PATHWAY Database (https://www.kegg.jp/kegg/pathway.html) [50], to classify the possible miRNA-related pathways involved in diseases and cellular events.

## 3. Results

### 3.1. Characteristics and Demographic Data

The average age of the patients with FCD was 37.4 ± 11.82 (mean ± standard deviation) years, and 37.7 ± 7.08 years in the control group. Based on the statistical analysis, age and sex were not significantly different between the two groups. All nine patients had refractory epilepsy to multiple AED therapy. Case #7 had surgery for epilepsy in 2008. However, the surgery was ineffective, and the epilepsy was still refractory to multiple AEDs. Histopathological tissue showed FCD type IIa, characterized by the loss of lamination of the cortex with disorganized medium to large-sized dysmorphic neurons. The clinical characteristics and demographic data of the nine patients with FCD are shown in Table 1.

### 3.2. Characterization of Exosomes from Human Serum

TEM images demonstrate the features of translucent, typically cup-shaped vesicles, corresponding to the characteristics of the exosomes (Figure 1a). We used Zetasizer to determine nanoparticles’ size distribution and the physical properties of the exosomes. The results showed that our serum exosome size was an average of 139.4 nm and the size distribution was in the range of 50 nm to 458 nm. The PdI value represents the size range of particles in the solution. The PdI value for exosomes in this study was 0.189, indicating a slightly broad particle size distribution due to the natural attributes of the exosomes being extracted from serum (Figure 1b).

Moreover, by Western blotting analysis, both exosomal markers including TSG 101, HSP70, and CD63 (Figure 2a), and neuronal markers including MAG [41], NCAM-1 [43], and GluR 2&3 [42] (Figure 2b) were detected in the circulating serum exosomes obtained from the control group. Our results demonstrated that we were able to obtain circulating exosomes from the peripheral blood samples (Figure 2). The isolated circulating exosomes are partially CNS-derived, as confirmed by neuronal markers (Figure 2b).

### 3.3. Expressional Changes of MicroRNA in Serum Exosomes from Patients with Focal Cortical Dysplasia

RNA-sequencing data generated with NGS were supplemented by bioinformatic analyses to identify potentially significant changes in the serum exosomal miRNA profiles in patients with FCD and controls. In order to clarify whether the circulating miRNAs in exosomes were differentially expressed in FCD, we performed a comparison analysis of miRNA modulation between patients with FCD and the controls.

Figure 3 showed differentially expressed miRNAs in circulating exosomes in patients with FCD versus controls. The expression of miRNA with >2-fold changes from NGS results was selected for further analysis. We obtained a total of 107 miRNAs with dominant fold change (>2-fold) from both the 5p-arm (Figure 3a) and the 3p-arm (Figure 3b) of 2780 mature miRNAs.

Among the 107 selected miRNAs, 46 5p-miRNAs and 61 3p-miRNAs were significantly upregulated in patients with FCD compared with the controls. Otherwise, there was no significantly downregulated expression among the 107 miRNAs. According to the differentially expressed profiles, 46 miRNAs were overexpressed (log_2_ value from 1.030 to 5.445) in the 5p-miRNA of FCD patients compared with the controls. In addition, 61 miRNAs were overexpressed (log_2_ value from 1.137 to 5.955) in the 3p-miRNA. These values were obtained from the calculated log_2_ fold-change between the two groups (Figure 3).

Hierarchical clustering analysis of 107 significantly expressed miRNAs in patients with FCD and controls was depicted with columns arranged by Heatmap MeV4.9 software analysis for NGS data (Figure 4). The sequential color scale differentiates high values (red) from low values (green) to represent the expression levels of each miRNA transcript. The color intensity of raw count data of 46 miRNAs was from 0 to 5.697 in the 5p-arm and the color intensity of raw count data of 61 miRNAs was from 0 to 10.338 in the 3p-arm.

### 3.4. Identification of Target MicroRNAs Related to Focal Cortical Dysplasia and Epilepsy

Based on the initial analysis with integrated platform of HMDD v3.2, miRway DB, and DIANA-miRPath v3.0 online tools, and further confirmed by MiRBase analysis, four potential targets of miRNAs, including miR-194-2, miR-15a, miR-132, and miR-145, from serum exosomes were identified related to FCD, seizures, status epilepticus, or epilepsy. These four predicted exosomal miRNAs had significantly differential expression in patients FCD compared with the controls, which may imply the evidence of related pathogenicity in patients with FCD. All four miRNAs had significantly differentially upregulated expression in serum exosomes between patients with FCD and controls. We performed a manual literature search in PubMed and the Human MicroRNA Disease Database (HMDD; v3.2) with the combined keywords “FCD, seizures, epilepsy or status epilepticus” and the four upregulated miRNAs (miR-194-2, miR-15a, miR-132, and miR-145), to confirm the relationship between the miRNAs and FCD or different types of seizures and epilepsy [52] (Table 2). According to the reported literature in PubMed, these miRNAs related to FCD and different forms of seizures and epilepsy [30,31,53,54,55,56,57] were identified, including the reported evidence of diagnostic and prognostic biomarkers and therapeutic targets (Table 2). Each single miRNA can influence multiple genes. The gene bioinformatics, regulated by four potentially predicted miRNAs in patients with FCD, were analyzed by HMDD V3.2 and are shown in Table 3.

### 3.5. Kyoto Encyclopedia of Genes and Genomes (KEGG) Pathway Enrichment Analysis and Pathway Category

A single miRNA can influence multiple genes and its related signaling molecular pathways and networks. Using the four upregulated miRNAs (miR-194-2, miR-15a, miR-132, and miR-145), we performed a KEGG pathway analysis to identify the possible signaling pathways related to FCD and refractory epilepsy. There were 20 enriched KEGG pathways and pathway target genes (Table 4) detected by the DIANA-miRPath v3.0 online tool. The functional category of pathways was determined using two integrated platforms, BioPlanet and KEGG PATHWAY Database, to classify the possible miRNA-related pathways category. The results showed that the pathway categories were related to the nervous system, cancer, cellular community, and cell growth and death (Table 4).

### 3.6. Related Pathways in Patients with Focal Cortical Dysplasia

Based on the results of HMDD V3.2 and the KEGG pathway analysis, we noted that the four predicted target miRNAs regulate gene expression and involve many pathophysiological pathways. Therefore, we selected the important target genes that are regulated by multiple miRNAs (≥2 miRNAs) to find related pathophysiological pathways in patients with FCD. We found that the expression of *CDKN1A*, *REL*, *CCND1*, *CDK6*, *FGF2*, *SOX5*, *BDNF*, *SK1*, *BMI1*, and *TP53* was regulated by two or more of these four predicted miRNAs (Figure 5). Therefore, eight possible related pathways were revealed by KEGG analysis (Figure 5). Among these eight predicted pathways, we indicated the five risk pathways involving more than three genes, regulated by four predicted miRNAs that included the mTOR signaling pathway, the PI3K-Akt signaling pathway, the p53 signaling pathway, cell cycle regulation, and the TGF-beta signaling pathway.

## 4. Discussion

miRNAs are the master regulators of gene expression. A single miRNA can influence multiple genes and proteins within diverse molecular pathways and networks [25,58]. Advanced studies have shown that miRNA may be the key in the pathogenesis of epilepsy [59]. miRNAs that target gene regulation may provide new challenges in the pathogenesis, diagnosis, and treatment of epilepsy and the translation into clinical practice [23,25,30,60]. However, the potential for miRNA-based therapeutics or diagnosis in epilepsy has not been indicated. Based on animal and human studies, many miRNAs have been suggested to play an important role in epilepsy and status epilepticus, including miR-23b, miR-15a, miR-132, miR-134, miR-145 miR-146a, miR-219, miR-199a, miR-128, miR-187, miR-124, miR-532, miR-365, miR-663b, miR-137, etc. [25,26,27,28,29,30,31,61]. These miRNAs in circulation in patients with epilepsy or animals have been reported to undergo upregulation or downregulation in different types of epilepsy, stage of seizures, and etiologies. Also, conflicting results may be present in different study models. The study indicated that the downregulated expression of miR-134, miR-181a, miR-15a, miR-194, and miR-106 could be a noninvasive diagnostic biomarkers for epilepsy patients [25]. Moreover, miR-124, miR-199a, and miR-128 could be candidate biomarkers in epilepsy diagnosis [25]. Several miRNAs, including miR-194, miR-301a, miR-30b, miR-342, and miR-4446, have been shown to be differentially regulated between the drug-responsive group and the drug-resistant group [58]. Whereas some circulating miRNAs have been suggested as diagnostic biomarkers, predictions of prognosis, or therapeutic targets in epilepsy [25,26,28,29,30,31,61], the potential roles of these miRNAs in epilepsy therapy and epileptogenesis, particularly in patients with FCD, remain unclear.

We noted that both exosomal markers, such as TSG 101, HSP70, and CD63, and neuronal markers, such as MAG, NCAM-1, and GluR 2&3, have been detected in circulating serum exosomes. We were able to obtain circulating exosomes from peripheral blood; a significant number of them were from CNS. Based on the analysis of exosomal miRNAs, we noted the upregulation of four putative miRNAs extracted from circulating serum exosomes in patients with FCD: miR194-2, miR15a, miR-132, and miR-145. According to the literature, these four miRNAs may relate to different forms of seizures, epilepsy, and status epilepticus. Among them, miR-132 was the most reported to be related to status epilepticus [31,62] and temporal lobe epilepsy [55,56]. In an animal study, upregulated expression of miR132 was noted in the hippocampus during status epilepticus [31]. The microinjection of locked nucleic acid-modified oligonucleotides against miR-132 depleted hippocampal miR-132 levels and reduced seizure-induced neuronal death [31]. In addition, miR-132 was shown to have increased expression in the human and rat epileptogenic hippocampus, particularly in glial cells [55,56]. These studies support roles for miRNAs, particularly miR-132, in the pathophysiology of status epilepticus. miRNAs may represent novel therapeutic targets to reduce brain injury and epileptogenesis [31,62]. miR194-2 and miR-15a are notably downregulated in epilepsy patients, so they may be used as novel biomarkers for the improved diagnosis and prognostic prediction in epilepsy [53,54]. In a human study, miR-145 was noted to be hypo-expressed in the surgical hippocampal tissues, but hyperexpressed in the blood of patients with mesial temporal lobe sclerosis [57]. However, another study [30] showed that miR-145 was significantly downregulated in the peripheral blood of patients with mesial temporal lobe epilepsy.

Malformations of cortical development (MCD) compose a diverse range of disorders that are common causes of neurodevelopmental delay and possible related refractory epilepsy [2]. FCDs are a related group of disorders with more localized dysplastic neurons that cause epilepsy and even refractory epilepsy [1,63]. Emerging evidence of molecular and genetic expression indicates that many malformations of brain developments, including FCD and some MCDs, are associated with abnormal neuronal proliferation, and result from gene mutations affecting mTOR and related pathways that serve as a central regulator of growth and homeostasis of neurons [2,11]. However, histology, genetic analyses, the topography of the lesions, and imaging characteristics suggest that FCD types are likely different entities [1,3,4,9]. Moreover, FCD is always a sporadic disorder without defined family pedigrees, and its pathogenesis remains unknown. Thus, some proposed potential pathogenic mechanisms besides mTOR pathways, including somatic gene mutation or a toxic insult to the developing brain, have been suggested [1,3,4]. The research for expressional changes in miRNAs is limited. In immature rats with liquid nitrogen lesion-induced FCD and cultured PC12 cells, miR-139-5p may play a crucial role in the modulation of cortical neuronal migration [64]. The differential expression of hsa-miR-4521 in the brain tissue and serum of refractory epilepsy patients suggested that serum hsa-miR-4521 may represent a potential diagnostic biomarker for FCD with refractory epilepsy [65]. The expression of miR-323a-5p was positively correlated with the duration of epilepsy and seizure frequency in patients with FCD [66]. These results suggest that the expression of miRNAs could be useful for improving diagnosis and monitoring treatment responses in patients with FCD. However, in the present study we did note see changes in miR-4521, miR-323a, or miR-139 expression in our patients with FCD. Further studies may be needed to elucidate the role of miR-4521, miR-323a, and miR-139 in FCD and refractory epilepsy.

In the present study, we also noted the upregulation of miR-132, together with three other miRNAs (miR194-2, miR15a, and miR-145) that downregulated some important genes’ expression, including *CDKN1A*, *REL*, *CCND1*, *CDK6*, *FGF2*, *SOX5*, *BDNF*, *SK1*, *BMI1*, and *TP53*. Considering genes’ interaction in networks, the inhibition of these important genes’ expression may involve some risk pathways that contributed to pathogenesis and epileptogenesis in patients with FCD and refractory epilepsy. Our results revealed eight high-risk pathways. Among these eight predicted pathways, we pinpointed five major high-risk of pathways: the mTOR signaling pathway, the PI3K-Akt signaling pathway, the p53 signaling pathway, cell cycle regulation, and the TGF-beta signaling pathway, involving three genes regulated by four downregulated miRNAs. We proposed that these five pathways may play crucial roles in pathogenesis and epileptogenesis in patients with FCD and refractory epilepsy.

Gene mutations in the mTOR signaling pathway have been well established, contributing to the pathophysiology of FCD and epileptogenesis [1,11,67,68]. Mutations in the *mTOR* gene and its related genes may lead to FCD and epileptogenesis in brain developmental malformations by an as of yet undefined mechanism [1,11,67,68]. The mTOR pathway contains serine/threonine protein kinase (a PI3K-related kinase), which brings together different extracellular stimuli, such as growth and nutrition factors, and diverges into several biochemical and molecular pathways [67,68]. Evidence suggests that some neurodevelopmental disorders are caused by somatic mutations in genes of the PI3K-AKT3 pathway. It was hypothesized that FCD might also be due to somatic mutations in genes belonging to the mTOR signaling cascade [68]. PI3K- and Akt-dependent mTOR activation has been reported in a hippocampal culture model of posttraumatic epilepsy in rats, and inhibition of PI3K, mTOR, or both prevented ictal activity and cell death [69]. Moreover, PI3K/AKT pathway mutations are an important pathophysiology of epileptogenic brain malformations, megalencephaly, hemimegalencephaly, and FCD [63]. Thus, our results also suggest that the mTOR signaling pathway and the PI3K-Akt signaling pathway are important risk pathways of pathogenesis and epileptogenesis in patients with FCD. This information also supports the fact that mTOR inhibitors, such as everolimus, may have therapeutic implications in the treatment of patients with FCD and refractory epilepsy [10].

The tumor suppressor p53 is a sequence-specific DNA binding protein that is primarily characterized as a transcription factor [70]; it regulates the expression of multiple target genes [71]. These genes are important regulators of glucose, lipid and amino acid metabolism, oxidative phosphorylation, growth factor signaling, reactive oxygen species (ROS) generation, mitochondrial function and integrity, and cell fate [71,72]. The p53-dependent cell death pathway was noted in perturbed progenitors and cortical malformations [73]. Interestingly, human specimens of FCD and animal models have demonstrated aberrant immunoexpressions of p53, which suggests that the p53 signaling pathway plays a crucial role in the pathogenesis of FCD [74]. Moreover, mTOR can induce apoptosis via the activation of p53 and the inhibition of the antiapoptotic protein Bcl-2 in response to cellular damage [75]. The induction of p53 expression may regulate the metabolism of neuronal cells differently, inducing autophagy and apoptosis, which are dependent on the regulation of the PI3K/AKT/mTOR pathway [71,76]. However, the mechanism by which the p53 signaling pathway controls cell growth via mTOR signaling is not yet firmly established. Our results also suggest that the p53 signaling pathway may regulate the mTOR signaling pathway and PI3K-Akt signaling pathway and act as an important pathogenic mechanism in FCD.

We also found that cell cycle regulation is a high-risk pathway in the development of FCD. Recently, in both rodent and primate studies, the regulation of the cell cycle, specifically of the G1 phase, was shown to play a crucial role in controlling area-specific rates of neuron production and the generation of cytoarchitectonic maps [10,77]. In general, brain development is associated with multiple molecular mechanisms, such as neurotransmitter release, cell cycle regulation, and cell‒cell communication [10,77]. In many aspects, the regulation of the cell cycle has been described as a key pathway in neuronal differentiation, proliferation, growth, and migration related to cortical development [10].

The TGF-beta signaling pathway is involved in a variety of biological processes during embryogenesis and in adult tissue homeostasis [78]. Recently, studies have explored the essential roles of TGF-beta signaling during neuronal development in the maintenance of neuronal activity [78]. Evidence implicates significant roles of the aberrant TGF-beta superfamily signaling in the pathogenesis of neurological disorders [78]. Cytokines/chemokines and the related TGF-beta signaling pathway have been demonstrated to play a role in patients with hippocampal sclerosis and FCD [79]. The TGF-beta signaling pathway in astrocytes play a critical role in post-injury epilepsy, so the manipulation of the TGF-beta pathway is a potential strategy for the prevention of post-injury epilepsy [80]. Although the role of the TGF-beta signaling pathway in FCD is unclear, in the present study we noted that the TGF-beta signaling pathway might be an important pathway involved in the pathogenesis of FCD. However, this needs further study before we can confirm it.

Recently, the contributions of somatic mutations and noncoding miRNAs have been explored in patients with epilepsy [24,25,58]. Further work is needed to explore the function of established epilepsy genes and exosomal miRNAs in targeting genes and protein regulation, their related signaling pathways, and crosstalk with epigenetics. These works may help with the translation to clinical care and the development of new AEDs and advanced therapeutic strategies in patients with FCD and refractory epilepsy.

## 5. Conclusions

In the present study, we found different expressions of circulating miRNAs extracted from exosomes in FCD patients, including miR194-2-5p, miR15a-5p, miR-132-3p, and miR-145-5p. Four miRNAs presented upregulated expression in patients with FCD compared with the controls. By KEGG analysis of four target miRNAs, we found eight possible signaling pathways that may be related to FCD and refractory epilepsy. We suggested that among them, the mTOR signaling pathway, the PI3K-Akt signaling pathway, the p53 signaling pathway, cell cycle regulation, and the TGF-beta signaling pathway are the high-risk pathways that play a crucial role in pathogenesis and epileptogenesis of FCD and refractory epilepsy. Our results suggest that circulating miRNAs from exosomes may provide a potential biomarker for diagnostic, prognostic, and therapeutic adjuncts in patients with FCD and refractory epilepsy.

## Figures and Tables

**Figure 1 cells-09-01867-f001:**
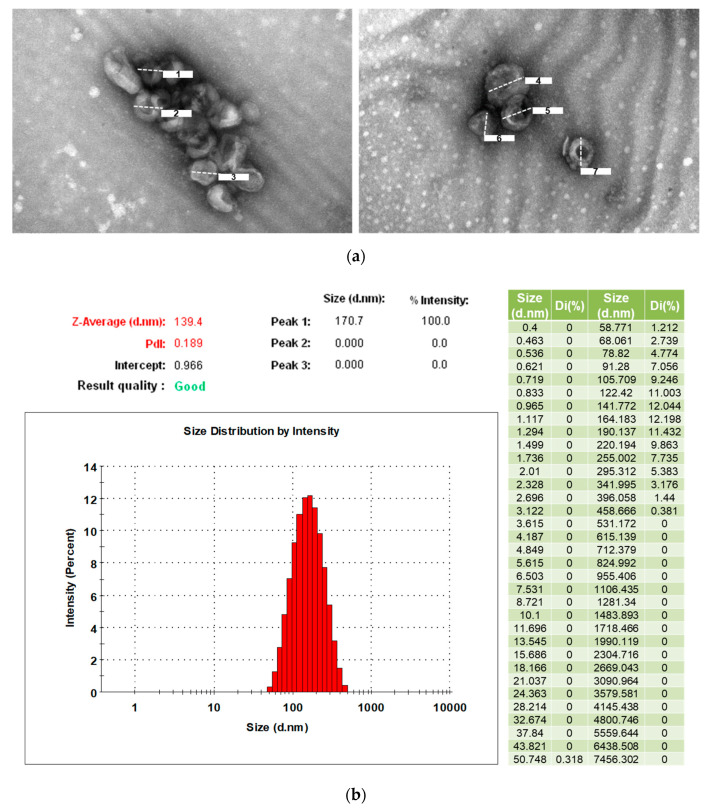
(**a**) Transmission electron microscopy (TEM) images demonstrate the characteristics of exosomes. Scale bars: (1) 79.3 nm, (2) 77.3 nm, (3) 92.4 nm, (4) 113 nm, (5) 92.3 nm, (6) 74.7 nm, (7) 80.6 nm. (**b**) The size distribution of the exosomes extracted from serum.

**Figure 2 cells-09-01867-f002:**
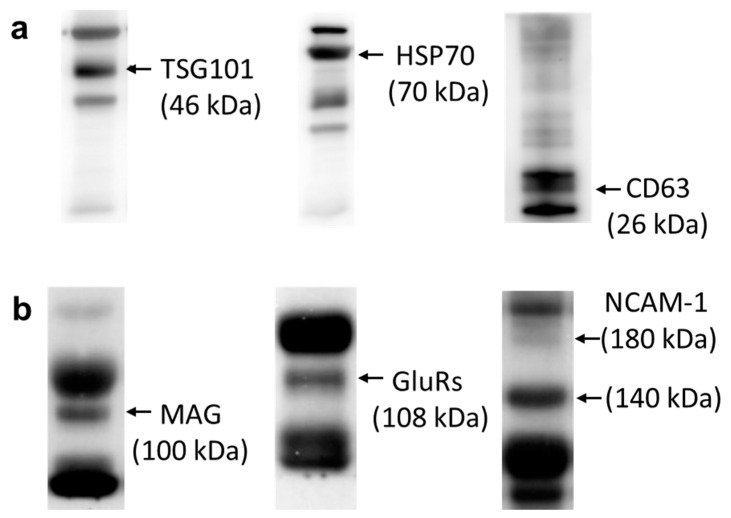
Western blotting analysis revealed that (**a**) exosomal markers TSG 101, HSP70, and CD63 were detected in the circulating serum exosomes. (**b**) Furthermore, the neuronal markers MAG, GluR 2&3, and NCAM-1 were also present in the serum exosomes from peripheral blood. CNS-derived exosomes were confirmed by neuronal markers.

**Figure 3 cells-09-01867-f003:**
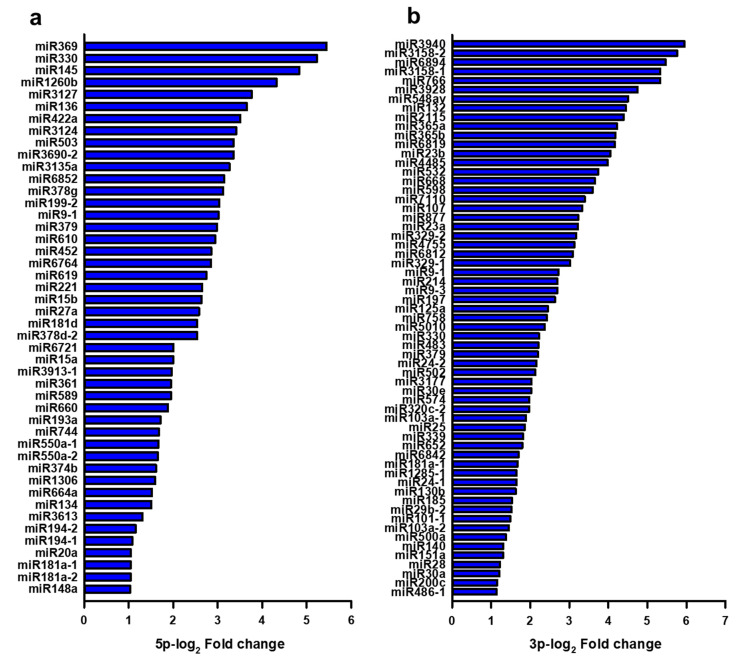
The expression profiles of miRNA extracted from serum exosomes were assessed using next-generation sequencing (NGS) analysis. The correlation analysis of microRNA (miRNA) expression in patients with focal cortical dysplasia and controls: 107 miRNAs showed statistically significant expression (*p* < 0.05), as assessed by a Student’s t-test. The bar chart indicates the 5p- (**a**) and 3p- (**b**) log_2_ ratio levels of mature miRNAs, sorted from highest to lowest in patients normalized to normal controls. log_2_ (mean of miRNA in patients/mean of miRNA in controls) expressed as log_2_ fold change.

**Figure 4 cells-09-01867-f004:**
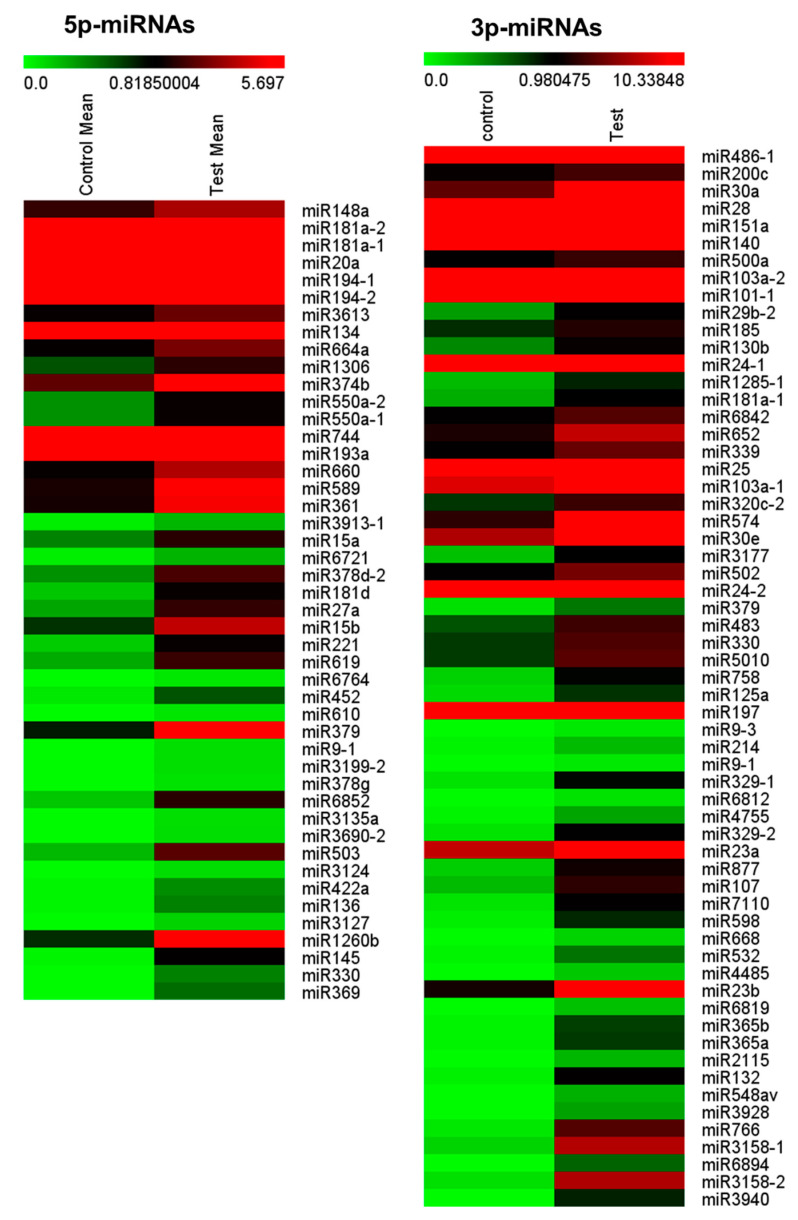
Cluster analysis of 107 differentially expressed miRNAs in patients with focal cortical dysplasia (Test Mean) and controls (Control Mean) by heatmap MeV4.9 software. Green = lower than mean intensity; red = higher than mean intensity. The conventional value of *p* < 0.05 was used for each miRNA.

**Figure 5 cells-09-01867-f005:**
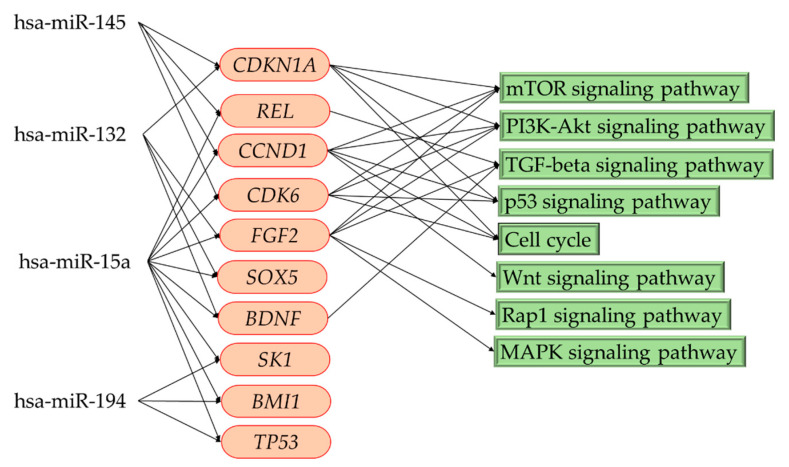
Multiple gene expressions regulated by four target miRNAs; a network model (KEGG) showed the possible pathophysiological pathways related to focal cortical dysplasia. has-miR: *Homo sapiens* microRNA.

**Table 1 cells-09-01867-t001:** Clinical characteristics and demographic data of patients with focal cortical dysplasia (FCD).

Case	Sex/Age	Seizure Semiology	Family History	MRI Finding	AEDs
1	F/39	Focal to bilateral tonic‒clonic seizure	-	Multiple bilateral subcortical heterotopia; right frontal FCD	CBZ, LTG
2	M/40	Focal onset clonic seizure	-	Bilateral frontoparietal and occipital heterotopia	LEV, ZNS, PER, LCM
3	M/29	Focal to bilateral automatic seizure	-	Right parietal FCD	OXC, PER
4	F/34	Focal impaired awareness clonic seizure	-	Right temporo-parietal FCD	LTG, LCM
5	M/36	Focal impaired awareness clonic seizure	-	Right frontoparietal FCD	LTG, PER, LCM
6	M/63	Focal impaired awareness dyscognitive seizure	-	Left frontoparietal lobe and subcortical band heterotopia	TPM, LEV
7	M/31	Focal to bilateral tonic‒clonic seizure	-	Left temporal lobe button of sulcus heterotopia	CBZ, LTG, PHT
8	F/18	Focal impaired awareness clonic seizure	+	Right frontoparietal lobe heterotopia	LEV, ZNS
9	F/47	Focal to bilateral tonic‒clonic seizure	-	Right posterior tempo-parietal and subcortical band heterotopia	VPA, LEV, LCM

MRI: brain magnetic resonance imaging; FCD: focal cortical dysplasia; AED: antiepileptic drug; CBZ: carbamazepine; LTG: lamotrigine; VPA: valproic acid; LEV: levetiracetam; ZNS: zonisamide; PER: perampanel; LCM: lacosamide; RFM: rufinamide; OXC: oxcarbazepine; TPM: topiramate; PHT: phenytoin.

**Table 2 cells-09-01867-t002:** Profiles of four identified target miRNAs in serum exosomes from patients with FCD.

miRNA ID	Stem‒Loop	Expression	Evidence	Disease Publication	PMID
miR-194-2	5p	Up	Diagnostic biomarker	Epilepsy	25825351
miR-15a	5p	Up	Diagnostic biomarker	Epilepsy Seizure onset and post-seizure	25825351 27840934
miR-132	3p	Up	Therapeutic target	Status epilepticus Temporal lobe epilepsy	21945804 31408236 24995086
miR-145	5p	Up	Circulating biomarker	Mesial temporal lobe epilepsy	27833019 31368064
MID: PubMed identification.					

**Table 3 cells-09-01867-t003:** Four target microRNAs regulating multiple genes.

miRNA	Regulation	Target Genes
hsa-miR-194	Down	*TP53*, *SKI*, *CARM1*, *BMI1*, *DNMT3A*, *CAV1*, *ICOSLG*, *IL10*, *YWHAE*, *SLC7A5*, *CPT1A*, *PEX26*, *AP1S1*
hsa-miR-15a	Down	*USP8*, *CCND2*, *CHD4*, *DIAPH1*, *BSG*, *BRCA1*, *ZBTB18*, *GNAL*, *RS1*, *SLC7A5*, *IRF4*, *CCND1*, *CDK6*, *RASSF5*, *CDC42SE2*, *APP*, *RBPJ*, *OCRL*, *BHLHE40*, *GCLM*, *MYB*, *BMI1*, *PEX13*, *ACOX1*, *EIF2B2*, *AKT3*, *EN2*, *SBNO1*, *TRAK1*, *CYP26B1*, *MBD4*, *WNK3*, *CHEK1*, *BCL2*, *HNRNPA1*, *NUFIP2*, *IKBKG*, *MTHFR*, *IFNG*, *BACE1*, *PAK2*, *AP3M1*, *BAP1*, *B3GNT2*, *CLCN3*, *PDE4D*, *RPS6KA3*, *ITGA2*, *NR2C2*, *RIMS3*, *HMGA1*, *CADM1*, *TMEM245*, *YAP1*, *CASK*, *KPNA3*, *WT1*, *CDKN2B*, *KIF1A*, *TLL1*, *L2HGDH*, *PAG1*, *DICER1*, *HNRNPDL*, *GRB2KMT2D*, *PDIK1L*, *PI4K2B*, *CTNNA3*, *SCAMP5*, *EFNB2*, *GSK3B*, *ARHGDIA*, *KIF5B*, *PNPO*, *MYO5A*, *REL*, *CA8*, *UBE2H*, *ALDH3B1*
hsa-miR-132	Down	*RAB18*, *ADGRF4*, *GDF5*, *FGF2*, *CHRNA5*, *NBN*, *NLGN2*, *S100A9 CHL1*, *MMP9*, *STMN1*, *IRAK1*, *CDKN1A*, *MAPK1*, *RGMB*, *NCS1*, *LIFR RTN4*, *SOX5*, *SOX6*, *POLK*, *BDNF*, *KPNA1*, *SIRT1*, *GPR153*, *SMN1*, *B3GAT1*
hsa-miR-145	Down	*SOD2*, *IFNB1*, *CD44*, *MDM2*, *TGFBR2*, *NR1D2*, *HDAC2*, *JAG1*, *GGCX*, *CTGF*, *NTRK2*, *MAP3K3*, *SERPINE1*, *CDKN1A*, *IRS2*, *DUSP6*, *CCND1 TPM1*, *CDK6*, *FXN*, *TGFB2*, *ACTB*, *BTG1*, *ITGB8*, *ESR1*, *REL*, *SMAD3 TPM3*, *CD40*, *SOX2*, *IGF1R*, *VEGFA*, *MUC1*

has-miR: *Homo sapiens* microRNA; the gene bioinformatics were analyzed by HMDD V3.2.

**Table 4 cells-09-01867-t004:** Kyoto Encyclopedia of Genes and Genomes (KEGG) pathway analysis and pathway category of four target microRNAs in serum exosomes from patients with focal cortical dysplasia.

Pathway Category	Gene Count	miRNAs	*p*-Value
**Nervous System**			
Glioma	25	4	0.00000
TGF-beta signaling pathway	27	4	0.00005
mTOR signaling pathway	25	4	0.00015
ErbB signaling pathway	28	4	0.00015
FoxO signaling pathway	41	4	0.00101
PI3K-Akt signaling pathway	79	4	0.00613
Hedgehog signaling pathway	18	4	0.01366
Axon guidance	30	4	0.03526
**Cancer**			
Proteoglycans in cancer	73	4	0.00000
Pathways in cancer	102	4	0.00000
Transcriptional misregulation in cancer	39	4	0.00101
**Cellular community**			
Signaling pathways regulating pluripotency of stem cells	53	4	0.00000
Focal adhesion	66	4	0.00000
Adherens junction	24	4	0.00011
**Cell growth and death**			
Hippo signaling pathway	52	4	0.00000
Wnt signaling pathway	40	4	0.00011
p53 signaling pathway	25	4	0.00048
Cell cycle	37	4	0.00311
MAPK signaling pathway	66	4	0.00483
Rap1 signaling pathway	51	4	0.01187

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
