# Peer review of "Circulating MicroRNAs from Serum Exosomes May Serve as a Putative Biomarker in the Diagnosis and Treatment of Patients with Focal Cortical Dysplasia"

_cells, 2020, doi:10.3390/cells9081867_

Round 1
Reviewer 1 Report
The authors investigated Circulating MicroRNAs From Serum Exosomes May Serve as A Putative Biomarker in Diagnosis and 4 Treatment of Patients with Focal Cortical Dysplasia The experimental techniques are straightforward, including transmission electron microscopy, western blotting, next-generation sequencing analysis
The investigation of novel biomarkers in epilepsy, including the field of miRNA, remains the focus for early diagnostification and treatment of epilepsy. Recent advanced studies have shown that miRNA may be a key in the pathogenesis and treatment of epilepsy. Taking this into consideration this article represents an important analysis in patients with FCD and some newly presented miRNA which are expressed more in this pathology playing role in epileptogenesis pathway. The results are clear and straightforward as far as they go. As indicated below, the authors could make some useful progress without a great deal of additional experimenting.
- Comments
Introduction and method:
- The authors have been focused on more general aspects in the introduction section. We suggest the authors focus more on the miRNA in the epilepsy-related studies, starting from preclinical evidence and the clinical studies which mentioned their potential in the diagnosis and treatment potential of epilepsy.
- The authors mentioned the regulation of miRNA and tried to link the exomes in this relation. We suggest the authors make a more strong connection between both related aspects since there is a fast and unclear dislocation of the topic from miRNA to the exomes and their role in the diagnosis.
- What is the implication of exome sequencing in other epilepsy types (check Demos M et al., 2019) with the respective diagnosing approaches in this relation. Authors should focus on these aspects with the related specific studies in exome and their role as a biomarker pool.
Method Section
- The authors need to send approval from the Clinical Study.
- Authors have used a lower number of patients to provide a biomarker indication. Which is the power of the sample analysis used in this study? We suggest the authors increased the number of patients.
- How did the authors select related antibodies? Authors need to describe more efficiently the western blotting technique, including mention of the second antibody as well, etc.
- Authors need to increase the evidence and description of NGS, moreover, how did the authors find the related biomarkers, pathogenicity, etc
- Why did the authors focus only on these databases?
- Which was the severity of the disease and the impact of the AED in the related expressions?
- Which are the MRI images from the analyzed group of patients?
- A complete list of patient characteristics needs to be included as a supplementary file?
- The relation of expressions of miRNA with the mTOR or TGF-beta signaling for instance is recommended.
Result and Discussion section:
- The WB bands are unclear, bands are separated and there is not any size presented within the bands. We suggest the authors clarify more the WB related Figure 2, dilution or antibodies, etc.
- The size presented in Figure 1 seems similar however the resolution of such figure is shown to be different for the right side
- Which are the statistical analysis obtained in Figure 3 with the respect of their miRNA findings, mean average and SD or SEM of related increase of expression, differences within the groups, etc?
- Did the authors indicate additional analysis of this expression to compare the time variables in this aspect? The authors need to specify the role of the diurnal cycle and all other additional factors which might have produced different results in such a small number of patients. Which was the time set of the sample intake?
- The authors need to explain the importance of expression and the following NGS in the presented result section? In table 2 the increased expression of such miRNA was shown and related to a biomarker or therapeutic target? Since these patients were in the treatment approaches how do the authors delineate better this finding? Which was the statistical comparison between the control patient without treatment or even healthy?
- Did the authors explore the related pathological SNPs which might deteriorate the expression?
- There are different literature approaches from Tiwari D et al., 2018; Yihong Ma et al., 2018 which need to be updated on the discussion part regarding the potential miRNA, epilepsy biomarkers, and the previously explored field related to miRNA concerning author findings. Authors need to compare other findings, including other related epilepsy pathologies as well.
- The traditional diagnostic methods mentioned in the discussion part should be transferred in the introduction, we suggest the authors refer more in their main related findings.
- Which are the implication of crosstalk with epigenetics in the relation of miRNA?
- The discussion of miRNA implicated signaling is always in a hypothetical manner rather than obtaining any sort of expression of the following signaling or related gene expression, proteins.
- The literature about FCD in the relation of miRNA is sparse. We suggest the authors compare their findings in this narrative as well.
Author Response
Response to Reviewer #1
Comments
Introduction and method:
- The authors have been focused on more general aspects in the introduction section. We suggest the authors focus more on the miRNA in the epilepsy-related studies, starting from preclinical evidence and the clinical studies which mentioned their potential in the diagnosis and treatment potential of epilepsy.
Response: Thank you for your suggestions. We have elucidated our introduction about preclinical and clinical evidence of miRNA and its relationship with diagnosis and treatment potential of epilepsy (Lines 83-89).
- The authors mentioned the regulation of miRNA and tried to link the exomes in this relation. We suggest the authors make a more strong connection between both related aspects since there is a fast and unclear dislocation of the topic from miRNA to the exomes and their role in the diagnosis.
Response: Thank you your comment. The role and importance of miRNAs in circulating CNS-derived exosomes have been elucidated our introduction (Lines 97-101).
- What is the implication of exome sequencing in other epilepsy types (check Demos M et al., 2019) with the respective diagnosing approaches in this relation. Authors should focus on these aspects with the related specific studies in exome and their role as a biomarker pool.
Response: In this study, we isolated the exosomes and extracted the miRNA in circulating miRNAs from circulating exosomes. The results WES of 9 patients is still unavailable and did not be analyzed in this study. However, we elaborated our “introduction section” about WES in patients with epilepsy (Lines 81-83) and added the Ref. Demos M et al., 2019.
Method Section
- The authors need to send approval from the Clinical Study.
Response: Approval from the Clinical Study has been sent to editorial office.
- Authors have used a lower number The of patients to provide a biomarker indication. Which is the power of the sample analysis used in this study? We suggest the authors increased the number of patients.
Response: Focal cortical dysplasia is not an uncommon disease. Moreover, we selected a small group of characteristic FCD by MRI study and clinical seizure presentations from all malformations of cortical development (MCD). We exclude other type or mixed type of FCD (Table 1.). Thus, the case number is lower.
- How did the authors select related antibodies? Authors need to describe more efficiently the western blotting technique, including mention of the second antibody as well, etc.
Response: the rationale of select antibodies, and method of western western blotting technique have been re-writing in the “Method: 2.5” (Lines 150-166).
- Authors need to increase the evidence and description of NGS, moreover, how did the authors find the related biomarkers, pathogenicity, etc
Response: the evidence and description of NGS have been elucidated in the “Methods” (Line 170-175).
- Why did the authors focus only on these databases?
Response: we selected at least six databases for analysis the significant miRNAs. These databases are common used and reliable in the literatures.
- Which was the severity of the disease and the impact of the AED in the related expressions?
Response: All patients had refractory epilepsy; their seizures are always severe. The AED treatment is complicated, it is difficult to analysis. The impact of AED may need the further large study.
- Which are the MRI images from the analyzed group of patients?
Response: the finding of MRI images was listed in Table 1.
- A complete list of patient characteristics needs to be included as a supplementary file?
Response: Clinical characteristics and demographic data of patients with FCD were listed in Table 1. We believe that presents clear and enough in clinical data.
- The relation of expressions of miRNA with the mTOR or TGF-beta signaling for instance is recommended.
Response: Thank you your comments. As your suggestion, we elucidated the miRNAs and possible risk signaling pathways in Lines 326 – 329.
Result and Discussion section:
- The WB bands are unclear, bands are separated and there is not any size presented within the bands. We suggest the authors clarify more the WB related Figure 2, dilution or antibodies, etc.
Response: We labeled the molecular seize in Figure 2. And the dilution of antibodies was listed in method section (Lines 150-166).
- The size presented in Figure 1 seems similar however the resolution of such figure is shown to be different for the right side
Response: Figure 1a contained two different EM images. Indeed, their resolution is different. To avoid confound, we have separated these two images.
- Which are the statistical analysis obtained in Figure 3 with the respect of their miRNA findings, mean average and SD or SEM of related increase of expression, differences within the groups, etc?
Response: we revised the Legend of Figure 3 (Lines 273-277), including the method of statistical analysis. The value is log2 (mean of miRNA in patients/mean of miRNA in controls) expressed as log2 fold change.
- Did the authors indicate additional analysis of this expression to compare the time variables in this aspect? The authors need to specify the role of the diurnal cycle and all other additional factors which might have produced different results in such a small number of patients. Which was the time set of the sample intake?
Response: To avoid other confounding factors, fasting peripheral blood was obtained from all subjects at 8:00 a.m. In patients with FCD, peripheral blood was obtained during an interictal state (Lines 120-121).
- The authors need to explain the importance of expression and the following NGS in the presented result section? In table 2 the increased expression of such miRNA was shown and related to a biomarker or therapeutic target? Since these patients were in the treatment approaches how do the authors delineate better this finding? Which was the statistical comparison between the control patient without treatment or even healthy?
Response: Thank you for your comments. We performed a manual literature search in PubMed and the Human MicroRNA Disease Database (HMDD; v3.2) with the combined keywords “FCD, seizures, epilepsy or status epilepticus” and the four upregulated miRNAs (miR-194-2, miR-15a, miR-132, and miR-145), to confirm the relationship between the miRNAs and FCD or different types of seizures and epilepsy. We have re-written and elucidated the Results (Lines 290-296).
- Did the authors explore the related pathological SNPs which might deteriorate the expression?
Response: Thank you for your suggestion. In this study, we did not investigate the related pathological SNPs. It may be further work.
- There are different literature approaches from Tiwari D et al., 2018; Yihong Ma et al., 2018 which need to be updated on the discussion part regarding the potential miRNA, epilepsy biomarkers, and the previously explored field related to miRNA concerning author findings. Authors need to compare other findings, including other related epilepsy pathologies as well.
Response: Thank you for your suggestion. We have added the discussion the different literature approaches from Tiwari D et al., 2018; Yihong Ma et al., 2018 (Lines 346-351)
- The traditional diagnostic methods mentioned in the discussion part should be transferred in the introduction, we suggest the authors refer more in their main related findings.
Response: As your suggestion, we transferred this sentence to “Introduction” (Lines 78-81).
- Which are the implication of crosstalk with epigenetics in the relation of miRNA?
Response: The evidence of crosstalk with epigenetics in the relation of miRNAs in not clear. We have elucidated our discussion in this point (Lines 459-464).
- The discussion of miRNA implicated signaling is always in a hypothetical manner rather than obtaining any sort of expression of the following signaling or related gene expression, proteins.
Response: Thank you for your suggestion. We have added this notion in our discussion (Lines 459-464).
- The literature about FCD in the relation of miRNA is sparse. We suggest the authors compare their findings in this narrative as well.
Response: The literatures of FCD and miRNAs is rare, we elucidated our discussion and added more information in Discussion (Lines 387-394).

Reviewer 2 Report
In this manuscript, the authors investigated the expression profiles of circulating exosomal miRNAs in patients with FCD and epilepsy. They identified 4 miRNAs that are upregulated in FCD patients. Their further pathway analysis of the 4 target miRNAs suggested that mTOR, PI3K-Akt, p53, cell cycle regulation and TGF-beta signaling pathways may be associated with pathogenesis and epileptogenesis in FCD. They concluded that circulating miRNAs from exosomes may serveas potential biomarkers for diagnosis and prognosis in patients with FCD and refractory epilepsy.
Overall, the manuscript was well written, although there are some grammatic errors and typos. The experiments were appropriately designed and the results are convincing. I recommend it for publication after better English editing.
Author Response
Response to Reviewer #2
Thank you for your comments. We have referred this manuscript to MDPI English Editing Service and Language has been corrected (English editing ID: English-8029).
Reviewer 3 Report
The authors present a nice study on the serum profile of miRNAs expression in a group of patients with malformation of cortical development compared to healthy controls. More in details, four overexpressed miR are detected, all known to be related to epilepsy or status epilepticus. Then authors report a very nice and detailed description of the genes regulated by these miR and the possible pathophysiological pathways involved.
Overall the paper is clear and results understandably discussed. I have only some suggestions/questions:
- Since the majority of these patients might have refractory epilepsy (almost all are treated with two AEDs), some of them could have undergone to epilepsy surgery. Do you have any information on the histology? Are the same mRNA profiles analysis been done also on brain specimen?
- It could be useful to add, as a control group, also a group of epileptic non FCD patients.
- I suggest to make a whole and complete english revision.
Author Response
Response to Reviewer #3
- Since the majority of these patients might have refractory epilepsy (almost all are treated with two AEDs), some of them could have undergone to epilepsy surgery. Do you have any information on the histology? Are the same mRNA profiles analysis been done also on brain specimen?
Response: Thank you for your comments. Case #7 had surgery for epilepsy in 2008. However, the surgery was ineffective, and the epilepsy was still refractory to multiple AEDs. Histopathological tissue showed FCD type IIa, characterized by loss of lamination of the cortex with disorganized medium to large-sized dysmorphic neurons (Lines 209-212). However, there was no research data of miRNAs in 2008.
- It could be useful to add, as a control group, also a group of epileptic non FCD patients.
Response: Thank your suggestion. Because the epilepsy is heterogenous including seizure types, severity, age, AED use, etc. Due to the budget in our study, we only selected a small group of characteristic FCD by MRI study and clinical seizure presentations from all malformations of cortical development (MCD). The further study may in other epilepsy patients without FCD may be further works.
- I suggest to make a whole and complete english revision.
Response: We have referred this manuscript to MDPI English Editing Service and Language has been corrected (English editing ID: English-8029).
Round 2
Reviewer 1 Report
Reviewers' Comments to Author:
Reviewer: 2
The author's reply queries raised in the first round of review and below are two minor additional comments:
- As I have pointed in the first round of revision, authors included the description in the NGS authors increased the evidence and description of NGS but they did not mention how they have selected such biomarkers or related pathogenicity.
- As I have pointed in the first round of revision, the WB bands are separated. We suggest the authors merge these bands in a single figure, moreover the GluRs protein seems very unclear and is very near to the bigger band (x? protein). Authors need to modify this or even put another band from other experiments (if the n sample size is adequate to better confirm this data).
Author Response
Dear Reviewer:
Thank you for your comments:
- As I have pointed in the first round of revision, authors included the description in the NGS authors increased the evidence and description of NGS but they did not mention how they have selected such biomarkers or related pathogenicity.
Response: The expression profiles of miRNA extracted from serum exosomes were assessed using next-generation sequencing (NGS) analysis. 107 miRNAs showed statistically significant expression (p < 0.05), as assessed by a Student’s t-test. For further confirmation, MiRBase, an open-source software library providing comprehensive prediction of miRNA targets, was used to predict the potential targets of significantly expressed (>2-fold change) miRNAs related to FCD, seizure, epilepsy or status epilepticus (Lines 192-194). These 4 miRNAs were predicted as a biomarker. These 4 predicted exosomal miRNAs have significantly differential expression in patients FCD compared with controls that may imply the evidence of related pathogenicity in patients with FCD and refractory epilepsy. We rewrote the Result section (Lines 288-298).
- As I have pointed in the first round of revision, the WB bands are separated. We suggest the authors merge these bands in a single figure, moreover the GluRs protein seems very unclear and is very near to the bigger band (x? protein). Authors need to modify this or even put another band from other experiments (if the n sample size is adequate to better confirm this data).
Response: The quality of western blotting images is not good. We revised the Figure 2., particularly GluRs. Also, the molecular weight of NCAM-1 is wrong. We have corrected it. Due to molecular weight of these protein is different, merging these band is difficult. We also provide the raw data to Reviewer for your references.
The western blotting was obtained from 4 controls and 2 patients.
We thus appreciate very much the opportunity to improve on our manuscript; and sincerely hope that our revision will now meet with your approval for publication in Cells.
Respectfully submitted,
